# Facile Preparation of a Bispherical Silver–Carbon Photocatalyst and Its Enhanced Degradation Efficiency of Methylene Blue, Rhodamine B, and Methyl Orange under UV Light

**DOI:** 10.3390/nano12223959

**Published:** 2022-11-10

**Authors:** Md. Akherul Islam, Jeasmin Akter, Insup Lee, Santu Shrestha, Anil Pandey, Narayan Gyawali, Md. Monir Hossain, Md. Abu Hanif, Se Gyu Jang, Jae Ryang Hahn

**Affiliations:** 1Department of Bioactive Material Sciences, Jeonbuk National University, Jeonju 54896, Korea; 2Department of Chemistry, Research Institute of Physics and Chemistry, Jeonbuk National University, Jeonju 54896, Korea; 3Functional Composite Materials Research Center, Institute of Advanced Composites Materials, Korea Institute of Science and Technology, Wanju, Jeonbuk 55324, Korea; 4Textile Engineering, Chemistry and Science, North Carolina State University 2401 Research Dr., Raleigh, NC 27695-8301, USA

**Keywords:** bispherical nanocomposite, photocatalysts, spherical silver nanoparticles, carbon nanospheres, reduced bandgap, Z-scheme process

## Abstract

The combination of organic and inorganic materials is attracting attention as a photocatalyst that promotes the decomposition of organic dyes. A facile thermal procedure has been proposed to produce spherical silver nanoparticles (AgNPs), carbon nanospheres (CNSs), and a bispherical AgNP–CNS nanocomposite. The AgNPs and CNSs were each synthesized from silver acetate and glucose via single- and two-step annealing processes under sealed conditions, respectively. The AgNP–CNS nanocomposite was synthesized by the thermolysis of a mixture of silver acetate and a mesophase, where the mesophase was formed by annealing glucose in a sealed vessel at 190 °C. The physicochemical features of the as-prepared nanoparticles and composite were evaluated using several analytical techniques, revealing (i) increased light absorption, (ii) a reduced bandgap, (iii) the presence of chemical interfacial heterojunctions, (iv) an increased specific surface area, and (v) favorable band-edge positions of the AgNP–CNS nanocomposite compared with those of the individual AgNP and CNS components. These characteristics led to the excellent photocatalytic efficacy of the AgNP–CNS nanocomposite for the decomposition of three pollutant dyes under ultraviolet (UV) radiation. In the AgNP–CNS nanocomposite, the light absorption and UV utilization capacity increased at more active sites. In addition, effective electron–hole separation at the heterojunction between the AgNPs and CNSs was possible under favorable band-edge conditions, resulting in the creation of reactive oxygen species. The decomposition rates of methylene blue were 95.2, 80.2, and 73.2% after 60 min in the presence of the AgNP–CNS nanocomposite, AgNPs, and CNSs, respectively. We also evaluated the photocatalytic degradation efficiency at various pH values and loadings (catalysts and dyes) with the AgNP–CNS nanocomposite. The AgNP–CNS nanocomposite was structurally rigid, resulting in 93.2% degradation of MB after five cycles of photocatalytic degradation.

## 1. Introduction

Numerous organic dyes contribute to water and environmental pollution. Organic dyes are endocrine-disrupting substances that can cause cancer, miscarriage, infertility, and reproductive problems. Therefore, the effective decomposition of organic pollutants is a crucial issue in water purification and in eradicating ecological pollutants. Recently, strong efforts have been made to develop more efficient organic and inorganic photocatalysts [1,2,3,4,5,6,7,8,9,10,11,12,13,14,15,16,17,18]. Ce-doped ZnO carbon nanofiber heterojunctions have received attention because carbon nanofibers synergistically increase photocatalytic activity [18]. However, many photocatalysts still have a wide bandgap [1,2,3,4], small specific surface area [4,5,6], low recycling stability [1,2,4,5,6,9,10,11], and/or a complex synthesis process [2,5,7,10,12,13,14,15]; in addition, many are secondary sources of environmental pollution [2,5,11,13,14,15]. These problems, coupled with the high-cost rare-earth materials, have greatly hindered their practical applications.

Carbonaceous materials are known to be effective catalysts in various chemical reactions. For example, biomass waste carbons have greater efficiencies in removing waste including heavy metals, pharmaceutical waste, pesticides, and microplastics [19]. However, their practicality as a photocatalyst to solve the wastewater problem is not yet certain, and their reaction mechanisms are unclear. Among the carbon used in various carbon nanostructures, carbon nanospheres (CNSs) can act as an electron trapper to increase conductivity, reduce charge recombination, and enable electron–hole (*e*^−^–*h^+^*) separation; it is therefore expected to be useful for an effective photocatalyst. However, CNSs have a relatively short photoexcited *e*^−^–*h^+^* pair lifetime, a poor specific surface area, a wide bandgap, and optical inefficiency, which are barriers to their widespread application as an effective photocatalyst for the CNSs alone [20]. There are also several drawbacks to the CNS preparation process. The authors of previous reports have used autoclave processes [1], chemical vapor deposition [2], arc plasma technology [3], and self-assembled template approaches [20] to produce different types of CNSs. These methods have disadvantages in that the size of the carbon nanospheres is large, large amounts of chemical reagents are required, and the synthesis process is complicated. In addition, there are few mono-component systems that meet all of the needs for an efficient photocatalyst. Therefore, various ways are needed to improve the catalytic efficacy of CNSs. One of these approaches is to construct nanocomposites between CNSs and other materials to promote charge transfer, improve the separation of charge carriers, and provide a more efficient bandgap arrangement.

Previous studies have revealed that Ag can be combined with other materials to construct photocatalysts. Examples include silver nanoparticles (AgNPs)–BiOCl [8], Ag/ZnO [21], Ag@CeO_2_ [14], Ag/BiVO_4_ [15], and Ag_2_CO_3_/Ag/AgNCO [12]. However, these methods have certain disadvantages such as a lower degradation efficiency because of these materials’ wide bandgaps, which reduces the absorption and utilization of UV light. Therefore, several groups have focused on preparing composite catalysts with improved properties such as a Ag@macroporous-resin [9], AgCl/Ag/AgFeO_2_ [4], β-Ag_2_MoO_4_/BiVO_4_ [5], Ag_3_PO_4_ [10], and AgI/Ag_2_CO_3_ [12]. Nonetheless, these materials demonstrate poor stability in reusability testing, exhibit long dye degradation times, involve large quantities of chemical reagents with solvents, or involve a complicated synthesis process [2,5,7,11,12,13,14,15]. Graphitic carbon nitride has been attracting much interest due to its good chemical stability, appropriate energy-band structure, and ability to be simply and inexpensively prepared from raw materials [22]. However, because of the high electrical resistance and limited active sites supported by the small specific surface area of C_3_N_4_, its use in photocatalysis has been limited [6]. Noble metals such as gold and silver can be useful components for photocatalysts. The ability to control the size and shape of noble metal nanoparticles has been well-developed. Thus, the surface area and the optical band can be controlled with noble metals. Au was combined with La to form a photocatalyst, where La ions trap the photogenerated electrons as an efficient scavenger, and Au has a higher specific surface, which is effective in increasing the photocatalytic activity [23,24]. Here, we developed an eco-friendly and facile method to produce AgNPs (~20 nm in diameter), CNSs (~20 nm in diameter), and bispherical AgNP–CNS composite particles (<50 nm in diameter) comprising AgNPs and CNSs. The bispherical AgNP–CNS nanocomposite was prepared in a sealed reactor at elevated temperatures from a mesophase of glucose and silver acetate. The prepared AgNP–CNS composite had a small particle size, high specific surface area, and large pore volume, facilitating the adsorption of organic pollutants onto its surface. The CNSs of the AgNP–CNS nanocomposite function as a charge-carrier bridge via covalent bonding at the heterojunction interface between the AgNPs and CNSs, resulting in a reduction in the bandgap. The band edges of the CNSs and AgNPs are well-aligned, leading to a *Z*-scheme to generate reactive oxygen species (ROS). Additionally, the photocatalytic efficacy of the AgNP–CNS nanocomposite was observed through the decomposition of organic pollutant dye under UV irradiation by maintaining various parameters such as acidity and dose loading (dyes and catalyst); the nanocomposite showed excellent performance within a short time compared with the other photocatalysts. In the reusability test, the AgNP–CNS nanocomposite displayed excellent catalytic activity, with only slightly reduced performance after five consecutive cycles.

## 2. Materials and Methods

### 2.1. Synthesis of AgNPs

AgNPs were synthesized via single-step heat treatment in a sealed vessel. Silver acetate (1.00 g, AgC_2_H_3_O_2_, 99%, Sigma-Aldrich, St. Louis, MO, USA) was loaded into a quartz crucible, which was subsequently located in a homemade stainless-steel chamber (SUS314) tightened with an oxygen-free high-conductivity (OFHC) Cu gasket to prevent air oxidation. The sealed chamber was heated in a furnace (KSL-1100X-S-UL-LD, MTI Corp., Richmond, CA, USA) for 5 h at 500 °C (Figure 1e,f). After completion of the heat treatment, the furnace was slowly cooled to room temperature to produce AgNPs with a mean diameter of 19.4 ± 6.0 nm.

### 2.2. Synthesis of CNSs

CNSs were prepared via the thermal carbonization of glucose, followed by thermolysis in a sealed vessel. Glucose (1.2 g, ≥99.5%, Sigma-Aldrich, St. Louis, MO, USA) was dissolved in distilled water (16.0 mL) with constant agitation at 700 rpm. The solution was located in a quartz crucible with a lid, and the loaded crucible was further covered with Teflon and located in a stainless-steel chamber sealed with an OFHC copper gasket. The sealed chamber was annealed to 190 °C at a rate of 6 °C min^−1^ and then kept at 190 °C for 24 h to form a mesophase (an intermediate phase, semi-carbonized, and dark-brown product, Figure 1). The mesophase was washed twice and purified to remove residual contaminants, as follows. First, the mesophase was cleansed with distilled water and centrifuged. Second, the centrifuged mesophase was washed with isopropanol and dried at 90 °C overnight. This purification process created uniformly distributed nanosized pores on the surface (increasing the specific surface area) and aided in the penetration of reactive species. The purified mesophase (0.5 g) was located in a quartz crucible with a lid, heated in the closed chamber to 500 °C at a rate of 10 °C min^−1^, and annealed for 6 h at 500 °C to form CNSs (Figure 1).

### 2.3. Synthesis of AgNP–CNS Nanocomposite

The AgNP–CNS nanocomposite (Figure 1c) was synthesized by a modified thermolysis process in a sealed vessel. Silver acetate was used as a precursor for the AgNPs in the AgNP–CNS nanocomposite (Figure 2a). The mesophase (0.50 g) and silver acetate (0.50 g) were mixed in a planetary centrifugal mixer (AR-100 mixer, Thinky, Laguna Hills, CA, USA) for 5 min to form a silver acetate layer on the surface of the particles of the brownish-black product (the mesophase, Figure 2a). The mixture was located in a quartz crucible with a lid, and the crucible was subsequently heated at 500 °C for 6 h in a stainless-steel reactor tightened with an OFHC Cu gasket to form a AgNP–CNS nanocomposite (Figure 2b). Annealing led to the simultaneous conversion of the silver acetate layer and mesophase into spherical AgNPs and CNSs with heterojunctions between them.

### 2.4. Physicochemical Properties of the AgNPs, CNSs, and AgNP–CNS Nanocomposites

Various techniques to analyze the physicochemical properties of the AgNPs, CNSs, and AgNP–CNS nanocomposites were employed and described in detail in Appendix A.

### 2.5. Photocatalytic Performance

The photocatalytic performance was assessed using organic pollutants: methylene blue (MB, Alfa Aesar, high purity, Heysham, UK), Rhodamine blue (RhB) (Sigma-Aldrich, St. Louis, MO, USA), and methyl orange (MO) (Sigma-Aldrich). Organic pollutants (1.75 mg in 0.100 L of distilled water) and the photocatalyst (20 mg) were mixed, and the mixture was magnetically agitated in a dark room for 1 h to stabilize the adsorption of the organic pollutant onto the photocatalyst surface (i.e., to reach an equilibrium state). The photodegradation evaluation was conducted under a UV lamp (wavelength 254 nm, FNS TECH, Chungnam, Korea). After a certain period of irradiation time, the sample (5 mL) was collected and a UV–Vis absorption spectrum was acquired to measure the degradation efficiency as a function of time. Similarly, AgNPs and CNSs were used as control catalysts to compare the photodecomposition of the MB, RhB, and MO pollutants. The degradation percentage was measured after UV illumination for 60 min in the presence of the AgNP–CNS nanocomposite, AgNPs, or CNSs.

#### 2.5.1. Point of Zero Charges (pH_PZC_) and pH

The pH for pH_PZC_ of the AgNP–CNS nanocomposite was obtained using the pH drift method. In each run, a volume of 0.01 M KCl solution was poured into a 150 mL beaker and the initial pH was controlled to the required level by the dropwise addition of 0.01 N NaOH or 0.01 N HCl solution. Then, 20 mL of the pH-controlled KCl solution and 20 mg of the AgNP–CNS composite were placed in a vial. The mixture was treated with a magnetic stirrer for 24 h to uniformly disperse the AgNP–CNS composite in the pH-controlled KCl solution. Initial pH levels of 2.00, 5.00, 9.00, and 12.00 were kept, and the corresponding final pH values were recorded after 24 h. The pH_PZC_ of the AgNP–CNS was calculated from a plot of the change in pH (ΔpH) as a function of the initial pH (pH_i_).

#### 2.5.2. Effect of the Loading Dose of MB and AgNP–CNS Nanocomposite on Photodegradation

Photocatalytic degradation was carried out using different amounts of dye (1.5, 2.0, 2.5, 2.5, and 2.5 mg MB) and AgNP–CNS nanocomposite (20, 20, 20, 25, and 30 mg) for 0, 5, 10, 20, 30, and 60 min under UV irradiation. MB degradation was evaluated in the presence of the AgNP–CNS nanocomposite.

### 2.6. Reusability Assessment

The recycling performance of the AgNP–CNS nanocomposite was evaluated. After photocatalytic degradation under UV light, the catalyst samples were centrifuged, filtered, and thoroughly cleansed with water. The samples were then dried at 90 °C for 60 min for reuse. The degradation percentage of MB was observed for five successive cycles, where each cycle lasted 60 min.

### 2.7. Charge-Carrier Trapping

Charge-carrier trappings were carried out to identify charge carriers and determine their photocatalytic function in MB decomposition. Three scavengers were used: potassium iodide (KI, Sigma-Aldrich, 99.99%), isopropyl alcohol (IPA, Sigma-Aldrich, 99.7%), and 1,4-benzoquinone (BQ, Sigma-Aldrich, 99.99%). IPA (3 mL), BQ (5 mg), and KI (5 mg) were applied to scavenge the photogenerated ^•^OH radicals, ^•^O_2_^−^ radicals, and *h*^+^, respectively.

### 2.8. Liquid Chromatography–Mass Spectrometry (LC–MS) Analysis

The LC–MS technique was utilized to detect fragments (intermediates) formed by MB degradation. Mass spectra of the MB decomposition products were recorded after 1 h of UV-light exposure in the presence of the AgNP–CNS nanocomposite. Each input included aliquots of 5 μL of nanocomposite injected into the LC system (LC 1200 Series, Agilent Technologies, Santa Clara, CA, USA). An electrospray ionization interface was utilized to produce ions in positive-ionization mode (+electrospray ionization).

## 3. Results and Discussion

### 3.1. Optical Characteristics of the AgNP–CNS Nanocomposite, AgNPs, and CNSs

UV–Vis spectra were measured to investigate the optical characteristics of the AgNPs, CNSs, and AgNP–CNS nanocomposite. Figure 1 shows the UV–Vis absorption scans of (a) AgNPs, (b) CNSs, and (c) the AgNP–CNS nanocomposite. The scans of the AgNPs and CNSs showed a maximum absorbance at 455 nm and a broad peak at 200–280 nm, respectively. The spectrum of the AgNP–CNS nanocomposite showed a broad peak within the range of 243–257 nm and an increase in absorbance at 580 nm. The peak at 580 nm may be due to a plasmonic resonance of AgNPs bonded to carbon nanospheres. Interestingly, compared with the AgNPs and CNSs, the AgNP–CNS nanocomposite showed increased absorbance over a wide range of UV and visible wavelengths. The absorption peak at 455 nm (Figure 1 (a)) was assigned to the surface plasmon resonance of Ag, indicating the formation of AgNPs. The broad absorption band at 200–280 nm was assigned to the π–π^∗^ transitions of C–C bonds of aromatic *sp*^2^ clusters [25] (Figure 1 (b)). The π–π^∗^ transitions can occur as a result of two different types of conjugation: nanometer-scale *sp*^2^ cluster formation and the connection of chromophore units (C=C, C=O, and C–O). In addition, in the scan of the AgNP–CNS nanocomposite, a broad absorption band appeared within the range of 243–257 nm, corresponding to the *n*–π* transition of CO bonds and the π–π^∗^ transition of carbon-containing groups (C=C, C=O, and C–O) of the CNSs in the AgNP–CNS nanocomposite. The different oxygen contents of the amorphous carbon in the CNSs and that in the AgNP–CNS nanocomposite might be related to the differences in the strong peak in the UV region of the corresponding scans. Notably, increased absorption over 240 nm for the AgNP–CNS nanocomposite can be advantageous for the development of UV-active photocatalysts.

We constructed Tauc lines to calculate the optical bandgap energies of the AgNPs, CNSs, and AgNP–CNS nanocomposite. The bandgaps for the AgNPs, CNSs, and AgNP–CNSs were determined to be 2.55, 2.90, and 2.52 eV, respectively (Figure 2a–c and Table 1). Remarkably, the bandgap of the AgNP–CNS nanocomposite was lower than that of the CNSs and AgNPs because of the formation of the nanocomposite. Previous reports have shown that the bandgap decreases upon adding Ag to carbon and ZnO. For example, the bandgaps of CNSs, a CNS/ZnO composite, and a Ag/ZnO/CNS composite were reported to be 3.54, 3.15, and 3.04 eV, respectively [25]. There may be an advantage to CNSs acting as a charge-carrier bridge at the heterojunction interface between AgNPs. Our AgNP–CNS nanocomposite had a smaller bandgap because the AgNPs chemically interact with the CNSs to form covalent bonds such as Ag–C or Ag–O–C. The smaller bandgap may enable the system to produce sufficient electron–hole pairs, which in turn generate robust reactive oxygen species.

### 3.2. Shape and Chemical Composition of the AgNPs, CNSs, and AgNP–CNS Nanocomposite

The crystallinity of the CNSs, AgNPs, and AgNP–CNS nanocomposite was investigated using their X-ray diffraction (XRD) plots (Figure 3). The XRD plots of the AgNPs and AgNP–CNS nanocomposite revealed peaks (2*θ*) at 38.15°, 44.32°, 64.50°, and 77.39°, which are associated with the (111), (200), (220), and (311) planes, respectively, corresponding to Ag in a face-centered cubic (*fcc*) crystal. The patterns of the AgNP–CNS nanocomposite and CNSs showed a broad carbon band at 24.82°, corresponding to the (002) plane of the graphitic structure. This broad band suggests that the carbon is poorly crystalline or amorphous. We also noted that the intensity of the carbon peak in the pattern of the AgNP–CNS nanocomposite was lower than that of the carbon peak in the pattern of the CNSs because of the interaction of the Ag particles with the CNSs. The sharp peak of Ag indicates that the Ag particles are crystalline.

We calculated the crystallite size of the products using the Scherrer equation [*D* = *kλ*/(*d* cos*θ*)], where *D* is the crystallite size, *k* is the Scherrer constant (0.9), *λ* is the X-ray wavelength used (0.154 nm), *d* is the full-width at half-maximum (FWHM) in radians, and *θ* is the diffraction angle in degrees. The average crystallite size was 34.75 ± 1.99 nm for the AgNPs, whereas that of the AgNP–CNS nanocomposite was reduced to 25.48 ± 1.46 nm. This decrease is attributed to the contribution of amorphous carbon in the nanocomposite. When only the four XRD peaks of the AgNPs in the AgNP–CNS nanocomposite were considered, the average crystallite size was calculated to be 36.23 ± 1.64 nm. This value was slightly greater than the average crystallite size of the bare AgNPs, possibly because of the longer annealing period for the nanocomposite compared with that for the bare AgNPs. A longer annealing time is advantageous because it provides sufficient time for a low-ordered phase to transform into a higher-ordered phase and consequently develop larger crystallites. The CNS crystallite size was 0.72 nm. The calculated crystallite size of the CNSs was small because of the contribution of amorphous carbon.

FTIR spectra were acquired in the wavenumber region 500–4000 cm^−1^ to identify the chemical groups present in the CNSs, AgNPs, and AgNP–CNS (Appendix A). Peaks at 1564 and 1403 cm^−1^ were recorded in the scans of all the samples. The peak at 1564 cm^−1^ was assigned to the asymmetric vibration of C=O and 1403 cm^−1^ to the vibration of a C–O group. The peak at 2371 cm^−1^ in the spectra of the CNSs and the AgNP–CNS nanocomposite was assigned to CO_2_. In the spectrum of the AgNPs, peaks were recorded at 3426, 1017, 922, and 649 cm^−1^. The broad peak at 3426 cm^−1^ was due to the presence of –OH groups, and the peaks at 649, 922, and 1017 cm^−1^ were due to Ag–O–Ag bonds in the AgNPs. The spectrum of the CNSs showed peaks at 3452, 1768, 1626, and 1265 cm^−1^, whereas that of the AgNP–CNS nanocomposite showed corresponding peaks at 3458, 1776, 1634, and 1275 cm^−1^, respectively. The broad peaks at 3452 and 3458 cm^−1^ were due to the presence of –OH groups on the CNSs and AgNP–CNS nanocomposite. The peaks at 1776 and 1768 cm^−1^ originated from the conjugated C=C stretching oscillations of conventional *sp*^2^-carbon in the AgNP–CNS nanocomposite and CNSs, respectively. The vibrational stretching of the carbonyl group of the nanometer-sized carbon was observed at 1634 and 1626 cm^−1^ in the scans of the AgNP–CNS nanocomposite and CNSs, respectively. The characteristic peaks at 1265 and 1275 cm^−1^ were ascribed to the stretching vibrations of the C–OH groups in the CNSs and the AgNP–CNS nanocomposite, respectively. Peaks were also observed at 2923, 2852, 864, 800, and 760 cm^−1^ in the scan of the AgNP–CNS nanocomposite. The peak at 2923 cm^−1^ arose from the asymmetric vibration of the *sp*^3^ bonds in CH_2_ and CH_3_ in the hydrocarbon network [26], and the peak at 2852 cm^−1^ was assigned to the overlapping of the symmetric vibration of the *sp*^3^ bonds of the CH_3_ and CH_2_ groups. Infrared absorptions at 760, 800, and 864 cm^−1^ were related to Ag–O–C bonding, confirming the formation of a heterojunction in the AgNP–CNS nanocomposite. Interestingly, the peaks at 3458, 1776, 1634, and 1275 cm^−1^ shifted slightly to higher wavenumbers in the spectrum of the nanocomposite compared with their positions in the spectrum of the CNSs, possibly because of the formation of heterojunctions in the composite.

We conducted XPS measurements to characterize the structure and functional groups of the AgNP–CNS nanocomposite. Figure 4a shows the survey spectrum of the AgNP–CNS nanocomposite, indicating the presence of Ag, C, and O. The high-resolution XPS scan showed two peaks (Ag-3*d*_3/2_ and Ag-3*d*_5/2_) at 367.7 and 373.7 eV, with a splitting of ~6 eV (Figure 4b), which indicates the presence of AgNPs [27]. Interestingly, the 3*d*_5/2_ peak of Ag appeared to shift to binding energies lower than the standard value (~368.2 eV for bulk Ag). This shift affirms the existence of interactions between Ag and carbon because the binding energy of Ag(I) is known to be lower than that of Ag(0) [28]. In addition, the binding energy is reduced because the electron density of Ag diminishes as a result of electron transfer from the AgNPs to carbon [29]. Figure 4c shows the C-1*s* peaks deconvoluted using the XPSPEAK41 software. The five peaks at 284.3, 284.6, 285.1, 286.39, and 288.4 eV are assigned to C=C (*sp*^2^) [30], C–C (*sp*^3^) [29], C–O (hydroxyl or epoxy) [31], C–O–C (ether) [32], and Ag–C–O–C–Ag, respectively [32] (Figure 4d). The peaks at 284.3, 286.39, and 288.4 eV further confirm the presence of a heterojunction with a strong binding between the AgNPs and CNSs via carbon and its functional groups. The presence of heterojunctions enhances the charge transfer and reduces the recombination of charge carriers.

Morphological details of the AgNP–CNS nanocomposite, AgNPs, and CNSs were obtained by the transmission electron microscopy (TEM) observation. The obtained spherical crystalline AgNPs are shown in Figure 5a. A histogram showing the distribution of the AgNP diameters was constructed using the ImageJ software. AgNPs with sizes ranging from 9 to 35 nm were counted and found to have a mean diameter of 19.4 ± 6.6 nm (Figure 5b). Figure 5c,d shows spherical carbon nanospheres with a size of 5–42 nm and an average diameter of 21.3 ± 9.7 nm. Appendix A displays the average diameter (19.34 ± 9.43 nm) of the AgNP-CNS nanocomposites. Low- and high-magnification results of the AgNP–CNS nanocomposite are displayed in Figure 6a,b. The images show the formation of multiple spherical CNSs and AgNPs in the AgNP–CNS nanocomposite and the formation of heterojunctions between the AgNPs and CNSs. The formation of these heterojunctions in the bispherical photocatalyst promotes charge carrier transfer between photoexcited constituents. Figure 6c shows a high-magnification image of the rectangular region in (b), where the crystallinity of the sample is evidenced by its lattice pattern. The lattice spacing of 0.23 nm was attributed to the *d*_111_ (=0.23 nm) spacing of *fcc* Ag [33], consistent with the XRD pattern of the nanocomposite (Figure 3).

The morphology and elemental composition of the AgNP–CNS nanocomposite, AgNPs, and CNSs were investigated by EDX analysis (Appendix A). The chemical composition was identified in the marked rectangular area in Appendix A. The EDX spectrum showed the C, O, and Si peaks for CNS, whereas the Ag, O, and Si peaks were observed for the AgNPs. Ag, C, O, and Si peaks appeared in the spectrum of the AgNP–CNS nanocomposite, without peaks attributable to other elements, indicating that the composite was produced by the combination of AgNPs and CNSs. The Si peaks originated from the sample holder.

BET/BJH analyses were performed to evaluate the *S*_BET_ and *V*_pore_ of the CNSs, AgNPs, and AgNP–CNS nanocomposite (Table 1). The *S*_BET_ of the AgNPs and CNSs was 9.10 and 8.24 m^2^g^−1^, respectively, whereas that of the AgNP–CNS nanocomposite was 15.67 m^2^g^−1^; thus, the *S*_BET_ of the composite was 172% and 190% greater than those of the AgNPs and CNSs, respectively. The *S*_BET_ of the AgNPs was slightly greater than that of the CNSs because the AgNPs exhibited a smaller average particle size. The average diameters of the AgNPs and CNSs were 19.40 ± 6.56 nm and 21.31 ± 9.74 nm, respectively (Figure 5). As noted, the *S*_BET_ of the AgNP–CNS composite was much greater than those of the CNSs and AgNPs because of the greater porosity of the structure and/or the presence of irregular interstitial pores in the composite. The pore volumes of the AgNP–CNS nanocomposite, AgNPs, and CNSs were 0.045, 0.027, and 0.020 cm^3^ g^−1^, respectively. The *V*_pore_ of the AgNP–CNS nanocomposite was also greater than those of the AgNPs and CNSs, which was attributed to gases formed during the heat treatment of the nanocomposite. In addition, longer annealing periods tend to result in larger specific surface areas. The extended surface area can provide more sites for the adsorption of reactive molecules, thereby enabling more efficient photocatalytic activity [30].

### 3.3. Thermal Properties of the AgNP–CNS Nanocomposite, AgNPs, and CNSs

The thermal properties of the CNSs, AgNPs, and AgNP–CNS nanocomposite were characterized by thermogravimetric analysis (TGA, Appendix A). For annealing, a maximum temperature of 800 °C, a rate of 10 °C min^−1^, and a N_2_ environment were used. CNS showed a slight mass loss starting at 100 °C because of the loss of moisture present in the sample. When the temperature reached 550 °C, a clear weight loss of ~95% of the original sample mass was observed. The AgNP–CNS nanocomposite and AgNPs started to decompose at ~360 °C. At 450 °C, the AgNP–CNS nanocomposite and AgNPs lost 35% and 48% of their respective masses. Both samples lost mass as the temperature was raised to 800 °C, where the remaining sample masses were 40.0% and 57.6% for the AgNPs and the AgNP–CNS nanocomposite, respectively, suggesting that the nanocomposite was thermally stable. The crystalline structure exhibited high heat resistance; thus, the decomposition percentage of a crystalline sample should be smaller than that of an amorphous phase. Therefore, the higher residual weight of the bispherical AgNP–CNS nanocomposite was attributed to its crystalline structure. Such high resistance to thermal decomposition also indicated a significant chemical interaction between the metal and nonmetal components of the composite [30].

### 3.4. Photocatalytic Performance of the AgNP–CNS Nanocomposite, AgNPs, and CNSs

The photocatalytic activity of the AgNPs, CNSs, and the AgNP–CNS nanocomposite was assessed on the basis of MB decomposition under UV light. Figure 7a displays the decomposition percentage over the illumination duration for MB in the presence of the CNSs, AgNPs, or AgNP–CNS nanocomposite. When the UV irradiation time was increased, the absorption values of the MB solution continuously decreased because the dye molecules were split into easily degradable molecules. The percentage of dye decomposition was determined according to the equation (*C*_0_ − *C*_t_)/*C*_0_ × 100, where *C*_0_ is the starting concentration of the MB solution before treatment and *C*_t_ is the concentration of MB after decomposition for period *t*. The results showed that only 73.2% and 80.2% of the MB were reduced in the presence of the CNSs and AgNPs, respectively, within 60 min, whereas 95.2% of the MB in the presence of the AgNP–CNS nanocomposite was decomposed within the same time period. These results show that the AgNP–CNS nanocomposite exhibited greatly improved photocatalytic performance compared with its AgNP and CNS components.

The photocatalytic efficacy of the AgNP–CNS nanocomposite was compared with the activity of the previously reported photocatalysts in Table 2. Various synthesis methods, dye concentrations, catalyst amounts, degradation times, and degradation rates were included in the summary, and the comparison confirmed that the AgNP–CNS nanocomposite developed in the present work exhibited excellent dye degradation performance (95%) over a period of 60 min. The excellent photocatalytic efficiency of the nanocomposite was attributable to the presence of heterojunctions between the AgNPs and CNSs. Such heterojunctions minimize the recombination rate of *e*^−^*–h^+^* pairs. As previously discussed, because the bandgap of the AgNP–CNS nanocomposite was significantly narrower than those of the AgNPs and CNSs, light absorption was increased. In addition, the *S*_BET_ of the AgNP–CNS nanocomposite could provide more catalytically active spots.

The photodegradation ratio (*C_t_*/*C*_0_) of MB dye vs. time is plotted in Figure 7b. With increasing exposure time, the dye percentage decreased almost linearly. The degradation kinetics were calculated according to the equation ln (*C*_0_/*C_t_*) = *kt* (Figure 7c), where *C*_0_ is the starting concentration, *C_t_* is the concentration at irradiation time interval *t*, and *k* is the rate constant (min^−1^). The calculated *k* was 0.025, 0.026, and 0.056 min^−1^ for the CNSs, AgNPs, and AgNP–CNS nanocomposite, respectively (Figure 7d). The *k* of MB degradation in the presence of the AgNP–CNS was 224% and 215% higher than those in the presence of the CNSs and AgNPs, respectively. Furthermore, the rate constant of the AgNP–CNS was substantially greater than the rate constants reported for other photocatalysts (Table 2). The results indicate that AgNP–CNS exhibited the highest *k-*value (0.056 min^−1^) among the reported photocatalysts.

The degradation behavior of the AgNP–CNS nanocomposite toward MO and RhB dyes was also evaluated under similar experimental conditions. Appendix A shows the decomposition percentage over the exposure time in the presence of RhB and MO. Appendix A shows the photodegradation ratio (*C*_t_/*C*_0_) as a function of the illumination time. Within 60 min, 76.7% of MO was decomposed, whereas 83.2% of the RhB was decomposed. The linear relation of ln (*C*_0_/*C*_t_) vs. illumination time affirms the first-order reaction (Appendix A). The *k* of the RhB and MO decompositions were 0.0245 and 0.0174 min^−1^, respectively (Appendix A). The decomposition rate of RhB was higher than that of MO because of the stronger azo bond in MO.

### 3.5. Effect of pH and Point of Zero Charge (PZC)

The decomposition rate depends on the properties of the dye (i.e., whether the dye is anionic or cationic) and on the PZC of the catalyst. The pH of the samples influenced the solubility of the dye and the surface chemistry of the adsorbent. The pH-PZC represents the surface charge. The pH-PZC values of the AgNP–CNS nanocomposite were evaluated by the pH drift method using NaOH and HCl to regulate the pH of the solution. A plot of pH_i_ and ΔpH (= pH − pH_i_) was constructed, where pH_i_ and pH are the initial and final pH, respectively. The pH-PZC value was assessed by plotting ΔpH vs. pHi, which revealed the pH-PZC (5.45) of the AgNP–CNS nanocomposite (Appendix A). In the present work, the surface charge of the AgNP–CNS catalyst was positive (cation) and remained positive at a low pH of 5.45. However, as the pH increased above 5.45, this positive value became negative.

The influence of pH on the decomposition of MB in the presence of the AgNP–CNS nanocomposite was assessed at various pH levels (2, 5, 9, and 12) (Appendix A). The degradation efficiencies were 75.9% (60 min), 83.8% (60 min), 88.5% (55 min), and 93.9% (50 min) at pH 2, 5, 9, and 12, respectively. The photocatalytic performance increased with the increasing pH of the MB solution, and the reaction at pH 12 exhibited the highest degradation activity. The high performance of the AgNP–CNS nanocomposite at pH 12 was attributed to the negative charges on the particle surfaces. That is, MB is a cationic dye (positive charge) that can adsorb onto the surface of the highly negatively charged AgNP–CNS nanocomposite via strong electrostatic attraction. This electrostatic interaction was advantageous for enhancing the adsorptive properties, thereby improving the dye degradation efficiencies. In addition, a higher pH value led to a higher concentration of OH^−^ ions that can form ^•^OH radicals upon interacting with holes, leading to an improvement in the photocatalytic efficacy [25]. In contrast, at lower pH values (less than 5.45, positively charged surface: acidic medium), the photocatalytic decomposition decreased because of the electrostatic repulsion of the cationic dye molecules. We also conducted a kinetic study of MB degradation (Appendix A). The degradation was found to allow the first-order reaction. The average value of the constant *k* was 0.059, 0.044, 0.036, and 0.028 min^−1^ for the degradation reactions conducted at pH 12, 9, 5, and 2, respectively (Appendix A). The rate constant at pH 12 was 210% greater than that at pH 2.

### 3.6. Effect of MB and AgNP–CNS Loading

Photocatalytic degradation reactions were carried out using various amounts of dye (1.5, 2.0, 2.5, 2.5, and 2.5 mg MB) and catalyst (20, 20, 20, 25, and 30 mg of AgNP–CNS nanocomposite); the results are illustrated in Appendix A. Initially, the dose of MB was increased from 1.5 to 2.5 mg (1.5, 2.0, and 2.5 mg) for a fixed amount (20 mg) of the AgNP–CNS nanocomposite. The degradation rate was 96.9%, 86.9%, and 78.8% for 1.5, 2.0, and 2.5 mg of dye, respectively. The results showed the reduced degradation efficiency for the larger quantity of MB (2.5 mg), which was attributed to fewer active spots on the surface because of a greater number of immobilized dye molecules. The formation of reactive species such as ^•^OH radicals on the surface was decreased because the active spots can be filled by dye ions as the number of dye molecules increases [34]. Another reason for the lower degradation at higher dye concentrations is that the solution’s transparency is reduced. The MB dose (2.5 mg) was subsequently fixed and the catalyst amount was increased (25 and 30 mg AgNP–CNS nanocomposite). In this case, the degradation percentages were 85.3% and 95.3% for 25 and 30 mg of catalyst, respectively. When the maximum catalyst amount was used (30 mg), the degradation efficiency increased because of the increase in the number of catalyst particles, which provided additional active sites. This trend with respect to increasing catalyst amount shows that the number of active spots increases because of the porous structure of the AgNP–CNS. Increasing the amount of porous structure increases the amount of surface defects, possibly increasing the amount of high-energy active sites. The degradation percentage only increases to a certain level because an excess amount of added catalyst induces aggregation or increases the turbidity of the solution. As a result, light penetration into the solution is inhibited and reduces the degradation percentage.

The kinetic parameter of dye degradation was studied on the basis of the linear relation between ln (*C*_0_/*C*_t_) and exposure time (Appendix A). The rate constants’ average values were 0.061, 0.039, and 0.032 min^−1^ for 1.5, 2.0, and 2.5 mg of MB, respectively, in Appendix A. The rate-constant value for 1.5 mg was 190% higher than that for 2.5 mg (catalyst: 20 mg). However, the MB quantity (2.5 mg) was unchanged, and the doses of catalyst (25 and 30 mg AgNP–CNS nanocomposite) were increased, resulting in a *k* of 0.038 and 0.054 min^−1^ for 25 and 30 mg of catalyst, respectively. The results show that the rate constant for 30 mg of catalyst was higher (142%) than that for 20 mg of catalyst when 2.5 mg of dye was used.

### 3.7. LC–MS Analysis of the AgNP–CNS Nanocomposite

LC–MS experiments were conducted to characterize intermediates produced during the decomposition of MB. Appendix A shows the mass spectra recorded before and after photocatalysis under UV exposure for 60 min. Appendix A shows a strong peak at *m/z* = 284, consistent with the presence of MB in the solution. The peak of MB (*m/z* = 284) completely disappeared, and several peaks associated with the intermediate products were newly observed (Appendix A). The elementary structure of MB was broken into numerous compounds, and the ultimate decomposition products were CO_2_, H_2_O, NO_3_^−^, and SO_4_^2−^. Appendix A shows a pathway for MB degradation, as proposed on the basis of the intermediates obtained in the mass spectra. The results confirm the effectiveness of the AgNP–CNS nanocomposite photocatalyst for MB dye degradation.

### 3.8. Reusability and Stability of the AgNP–CNS Nanocomposite

Reusability and stability are critical factors for photocatalysts to be used in practical applications. Here, we evaluated the reusability and stability performance of the AgNP–CNS nanocomposite. The decomposition percent of MB was observed for five successive cycles, where each cycle was 60 min (Figure 8a–e). The catalyst samples were filtered and thoroughly cleaned with distilled water after each complete cycle, and a fresh MB solution was used in the subsequent cycle. The results show that the decomposition efficiency was 95.2% in the first cycle and remained 93.2% after five consecutive cycles (Figure 8f). The minor decrease in performance (2.01%) was due to a loss of the catalyst during recycling. Here, we calculated the rate constant for the five reusability cycles. The *k*’s were 0.0594, 0.0587, 0.0586, 0.05855, and 0.05850 min^−1^ for the five consecutive recycles. These results show that the rate constant for five cycles was not substantially reduced (Figure 8g).

The results of the reusability test were compared with the results for the previously reported photocatalysts (Table 3). Our AgNP–CNS exhibited greater stability during five recycles than the previously reported photocatalysts. Additionally, XRD measurements were carried out before and after five cycles to compare the structural changes of the AgNP–CNS (Figure 8h). All of the prominent peaks in the XRD plot of the AgNP–CNS nanocomposite were retained after five cycles. Thus, the AgNP–CNS nanocomposite demonstrated excellent reusability and stability for treating an organic dye under UV exposure.

### 3.9. Scavenger Experiments

Charge-carrier trappings were performed to characterize the most reactive species associated with the photodecomposition reactions and to probe the MB photodegradation process. The degradation of MB through AgNP–CNS was conducted using IPA, BQ, and KI as the capture agents of ^•^OH, ^•^O_2_^−^, and *h*^+^, respectively. Appendix A shows that the reaction process slowed when the scavengers were present. Among them, IPA slowed the degradation efficiency of the dye most extensively, implying that ^•^OH was the most active species. Based on the depressant-function order of IPA > BQ > KI, the species activity order was ^•^OH > ^•^O_2_^−^ > *h*^+^. Although ^•^O_2_^−^ was the second-most important component of the decomposition, the effect of an increase in the amount of *h*^+^ on the rate was also noticeable. All of the reactive species (^•^OH, ^•^O^2−^, *h*^+^) were produced and crucial for accelerating the photocatalytic decomposition. These findings are consistent with the proposed process of the photocatalytic degradation with AgNP–CNS.

### 3.10. Proposed Mechanism of the Photocatalysis

The photocatalytic decomposition of pollutants is associated with several parameters including the *S*_BET_, *V*_pore_, optical bandgap, crystallite size, and band-edge potential of the catalyst. According to the above analyses, the prepared AgNP–CNS nanocomposite showed a high *S*_BET_, narrow bandgap, and large crystallite size. Another critical factor governing photocatalytic performance is charge separation efficiency. Therefore, determining the value of the conduction band (CB) and valence band (VB) edge potentials of the AgNP–CNS nanocomposite is important. The CB and VB were determined using the following equations:*E*_CB_ = *χ* − *E*_e_− 0.5*E*_g_(1)
*E*_VB_ = *E*_CB_ + *E*_g_(2)
where *χ*, *E*_g_, and *E*_e_ represent the absolute electronegativity of the semiconductor, the bandgap energy of the semiconductor, and the energy of free electrons (4.5 eV on hydrogen scale), respectively. *E*_CB_ and *E*_VB_ are the CB and VB edge potentials, respectively. The computed values of *E*_CB_ for the CNSs and AgNPs were 0.34 and −1.47 eV, respectively. The *e*^−^_CB_ and *h*^+^_VB_ were produced on the CNSs under UV-light exposure. The photogenerated *e*^−^_CB_ migrated to the VB of the AgNPs and reduced the *e^−^*–*h^+^* recombination. The *E*_CB_ of the AgNPs was much lower than the potential of O_2_/^•^O_2_^−^ (−0.33 eV); therefore, dissolved oxygen molecules were modified to ^•^O_2_^−^ according to Equation (3):O_2_ + e^−^_CB_ → ^•^O_2_^−^(3)

The ^•^O_2_^−^ radicals generate H_2_O_2_ through several stages (Equations (3)–(8)):H^+^ + ^•^O_2_^−^ → HO_2_^•^(4)
H^+^ + HO_2_^−^ → H_2_O_2_
(5)
O_2_ + 2H^+^ → H_2_O_2_
(6)
HO_2_^•^ + HO_2_^•^ → O_2_ + H_2_O_2_(7)
HO_2_^•^ + ^•^O_2_^−^ → O_2_ + HO_2_^−^(8)

The H_2_O_2_ subsequently breaks down into ^•^OH radicals. Thus, the produced ROS (^•^OH, O_2_^•^, etc.) are critical for pollutant decomposition and function as major oxidants (Equations (9)–(11)):OH^−^ → ^•^OH (9)
H_2_O_2_ → 2 ^•^OH(10)
Pollutant + (*h*^+^, ^•^O_2_^−^, ^•^OH) → Decomposition products (11)

The calculated values of *E*_VB_ for the CNSs and AgNPs are 3.20 and 1.35 eV vs. the normal hydrogen electrode (NHE), respectively. The oxidation of OH^−^ or H_2_O to ^•^OH radicals through the photogenerated holes was possible because the *E*_VB_ band potential of the CNS was greater than the standard potential of the OH^−^/^•^OH redox couple (+2.40 V vs. NHE) [25]. The *h*_VB_^+^ potential of the CNSs and AgNPs was also energetically favorable for direct hole oxidation of the MB, MO, and RhB dyes. The computed values of the CB and VB were used to illustrate the proposed process of the photocatalytic reaction (Figure 9).

In addition, the potentials of the OH^−^/^•^OH and O_2_/^•^O_2_^−^ redox couples were +2.40 and −0.33 eV, respectively. Therefore, generating both active species may be impossible in the case of a mono-component catalyst of either AgNPs or CNSs. These findings support the role of the charge-carrier migration process (*Z*-scheme heterojunction) during photocatalysis action.

Our synthesis method is environmentally friendly and facile to produce AgNPs, CNSs, and bispherical AgNP–CNS composites with small sizes, which is advantageous in that the specific surface area of the AgNPs-CNSs nanocomposite is higher. In addition, purification of the mesophase can create uniformly distributed nano-sized pores on the surface and aid in the penetration of reactive species. As mentioned, the *V*_pore_ of the AgNP–CNS nanocomposite was greater than those of AgNPs and CNSs due to the gases formed during heat treatment of the nanocomposite.

## 4. Conclusions

We developed a facile procedure to prepare AgNPs, CNSs, and an AgNP–CNS nanocomposite and demonstrated the excellent photocatalytic performance of the nanocomposite. The prepared samples were synthesized by modified thermolysis processes in a sealed vessel. The dye degradation efficiency in the presence of the AgNP–CNS nanocomposite was 215% and 224% greater than the efficiencies of dye degradation in the presence of the AgNPs and CNSs, respectively. The narrow bandgap of the nanocomposite results in an enhanced ability to absorb and utilize UV light. The increased *S*_BET_ of the nanocomposite provided more active spots for the adsorption of reactive molecules. The AgNP–CNS nanocomposite also exhibited a larger crystallite size, which was a significant factor for the mass production of photogenerated radicals capable of rapidly degrading dye solutions. The VB and CB edge locations of the nanocomposite are advantageous for generating substantial concentrations of ^•^OH, ^•^O_2_^−^, and *h*^+^ and for efficiently separating charge carriers. Radical **˙**OH is the major ROS, followed by **˙**O_2_^−^ and *h*^+^. Photocatalytic MB degradation was evaluated at various pH levels and using various doses of the AgNP–CNS nanocomposite catalyst. The photocatalytic degradation efficiency of the AgNP–CNS nanocomposite was 83.2% for RhB and 76.7% for MO. MB was almost decomposed within 50 min at pH 12. The decomposition could only produce clumps or make the solution more turbid up to certain limits because of the addition of excess catalyst. The AgNP–CNS nanocomposite showed only a slight decrease in activity (~2%) after five consecutive cycles.

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
