# Peer review of "Facile Preparation of a Bispherical Silver–Carbon Photocatalyst and Its Enhanced Degradation Efficiency of Methylene Blue, Rhodamine B, and Methyl Orange under UV Light"

_nanomaterials, 2022, doi:10.3390/nano12223959_

Round 1
Reviewer 1 Report
In this manuscript, authors developed a facile preparation of the AgNP-CNS(silver nanoparticles carbon nanospheres) which showed excellent photocatalytic efficacy in dye degradation. AgNPs were successfully synthesized by single-step heat treatment and AgNP-CNS nanocomposite was synthesized by modified thermolysis process. After that, important indicators of dye degradation such as physicochemical properties, photocatalytic performance, reusability were tested. Actually, this manuscript is well written, AgNP-CNS nanocomposite has excellent performance in dye degradation as visualized by the rich characterization results data. However, there are still some issues to be addressed. This reviewer would suggest a moderate revision before its acceptance.
1. One sentence to provide the aim or background of this work should be provided at the beginning of abstract.
2. The background on the dye issue in water and environment pollution should be further clarified with some recent articles as support: Synthesis and Application of Granular Activated Carbon from Biomass Waste Materials for Water Treatment: A Review; Synthesis of lignin-poly(N-methylaniline)-reduced graphene oxide hydrogel for organic dye and lead ions removal; Chemical Engineering Journal 446, 136851, 2022; etc.
3. Additional information about the advantages of choosing Ag to combine with carbonaceous materials could enhance the persuasive power of the article. Recently, La and Au can be combined with other to construct photocatalysts. It could be better to briefly compare with other metals highlights the advantages of silver.
4. More background on the photodegradation of dyes should be provided with supporting articles: Chemical Research in Chinese Universities 37, 565–570, 2021;
5. The images of experimental samples should be showed in the article.
6. The UV–vis absorption scans of AgNPs, CNSs, and the AgNP–CNS nanocomposite could be combined and indicated with different colors.
7. It could be better to use as little black as possible for the lines in the figures.
8. The size of words in tables should be modified.
9. The facile and convenient characterization of the advantages of the synthesis method should be discussed in results and discussion.
10. Element mapping should be provided to determine the element distribution.
11. The title of article should be revised. There are more common dyes than just MB(methyl blue), and this piece of article on the degradation of dyes are all experiments with MB as an example. The degradation of MB should be pointed in title.
12. Authors should carefully check the format of the manuscript to ensure the uniform format is used all through the manuscript.
Author Response
We uploaded our responses as a file.

Reviewer 2 Report
This manuscript describes the development of a novel Ag-Carbonaceous nano-composite photocatalysts in order to degenerate dyes. The manuscript is clearly written, contains description of the processing and detailed analysis of the performance. A detailed analysis to literature data is provided, and documents the superior performance of the newly produced nano-composite. The English is easy to understand.
All figure captions and lines 203, 560 to 588: Line spacing is too large
Line 143 Scheme 2(a)
The graph shows a white layer surrounding the semi-carbonized carbon and a bluish layer, both have been allocated as silver acetate layer. Are they the same? Then they should be displayed in the same color. Or, what is the difference? Then, the difference should be emphasized in the text.
Scheme 2(b) What is the meaning of the small white dots on the Carbon nanosphere? Are Carbon nano-spheres inherent inhomogeneous?
Line 156 English correction
and a photocatalyst (20 mg) were mixed, -> and the photocatalyst (20 mg) were mixed, or: " the three photocatalysts considered in this study"
Line 244 (fig. 2) Apparently fig 2(b) and 2(c) have been displayed in the wrong order, needs correction.
Line 255 Carbon can be either crystalline or amorphous. What do the authors want to express with this sentence?
Line 355 in comparison to table 1: There is a discrepancy in the average crystalline size between TEM images and table 1 (AgNP 34.75 vs 19.4 nm, and AgNP-CNS 0.72 vs 21.31 nm). Please explain!
The size distribution of nano-spheres of AgNP-CNS (fig.6) is not shown.
Line 392: What kind of light source were used for UV irradiation? Please add this important information.
Supplement data: Fig S8 (b) The yellow line is drawn, but there are no data points (except at 0,0). Please correct.
Author Response
We uploaded our responses as a file.

Round 2
Reviewer 1 Report
Authors have addressed all the issues well except one minor problem. The author name should be rechecked, especially the refences with red color. The surname and the given name are reversed the order, especially the Chinese names.
Author Response
We thank you for the correction and we are very sorry for our inadvertent misnaming of references authors. References have been carefully checked and corrected.